# Ultrafast valence to non-valence excited state dynamics in a common anionic chromophore

James N. Bull [1], Cate S. Anstöter [2] & Jan R.R. Verlet [2]*

Non-valence states in neutral molecules (Rydberg states) have well-established roles and importance in photochemistry, however, considerably less is known about the role of non-valence states in photo-induced processes in anions. Here, femtosecond time-resolved photoelectron imaging is used to show that photoexcitation of the $S_1(\pi\pi^*)$ state of the methyl ester of deprotonated *para*-coumaric acid – a model chromophore for photoactive yellow protein (PYP) – leads to a bifurcation of the excited state wavepacket. One part remains on the $S_1(\pi\pi^*)$ state forming a twisted intermediate, whilst a second part leads to the formation of a non-valence (dipole-bound) state. Both populations eventually decay independently by vibrational autodetachment. Valence-to-non-valence internal conversion has hitherto not been observed in the intramolecular photophysics of an isolated anion, raising questions into how common such processes might be, given that many anionic chromophores have bright valence states near the detachment threshold.

[1] School of Chemistry, Norwich Research Park, University of East Anglia, Norwich NR4 7TJ, UK. [2] Department of Chemistry, Durham University, South Road, Durham DH1 3LE, UK. *email: j.r.r.verlet@durham.ac.uk

The active site in photoactive proteins and biomolecules is often a simple, well-defined anionic chromophore[1,2]. For example, one of the most studied biochromophores is that found in the photoactive yellow protein (PYP)[3], where an anionic thioester derivative (Fig. 1a) of the E-isomer of para-coumaric acid undergoes an ultrafast $E \rightarrow Z$ isomerisation that triggers a conformational change in the protein and ultimately leads to a phototaxis response of the host bacterium[4–9]. One of the primary goals of protein photochemistry is to understand the properties of the chromophore and how a protein or condensed phase environment influences the photochemistry. In this context, it is interesting to study biochromophores in isolation, i.e., in the gas-phase, where measurements directly probe the intrinsic light-driven response of the chromophore and also provide theoretically tractable benchmarks[10–12]. Interest in the inherent photochemistry of model PYP chromophores has prompted numerous gas-phase action spectroscopy[13–17] and

photoelectron spectroscopy studies[18–21], mostly considering deprotonated para-coumaric acid and ester derivatives. However, these studies utilised light sources with nanosecond pulse duration that are not suitable for characterising the ultrafast excited state dynamics that define the photophysics of the PYP chromophore. Furthermore, difficulties in the interpretation of the gas-phase spectra is compounded by the fact that the photoactive $S_1(\pi\pi^*)$ excited state is close in energy to the electron detachment threshold. The consequence of this near degeneracy is that electron loss by autodetachment can compete with excited state nuclear dynamics (isomerisation). Such a competition was evident in the only previous time-resolved study on an isolated deprotonated para-coumaric acid derivative containing the methyl ketone functional group[22]. Moreover, because of the proximity of the $S_1(\pi\pi^*)$ state to the detachment threshold in most PYP chromophore models, the $S_1(\pi\pi^*)$ state could potentially couple to non-valence states of the anion. Specifically, the neutral molecular core of the model PYP chromophore anions in all studies to date have dipole moments that exceed the ~2.5 D threshold to support a dipole-bound state (DBS)[23,24], and therefore will have at least one DBS. While DBSs have been observed in many action and photoelectron spectra by direct excitation[23,25,26], internal conversion from a valence-localised state to a DBS has only been observed directly in cluster systems[27]. In the context of fundamental chemical physics, the lack evidence for the participation of non-valence states in the excited state dynamics of anions is surprising considering that nonadiabatic dynamics between valence and Rydberg states is a common process in the photochemistry of neutral molecules[28,29]. For PYP chromophore dynamics, the participation of non-valence states has not been considered.

Disentangling environmental perturbations on the excited state dynamics of model PYP chromophores has led to some uncertainty and controversy. There are open questions regarding whether $E \rightarrow Z$ photoisomerisation actually occurs for many of the model PYP chromophores in the gas-phase and how solvation might influence the photoisomerisation dynamics[17]. For example, studies on para-coumaric acid derivatives in solution have suggested that the $E \rightarrow Z$ photoisomerisation efficiency and timescale are modified by a number of factors, including the chromophore's deprotonation state; functional group substitution on the carbonyl group or torsion-locking the single-bond linking the carboxylic acid tail to the phenyl ring; and properties of the solvent such as polarity and viscosity[30–36]. Theoretical studies have confirmed that solvation and substitution on the carboxylate group of para-coumaric acid derivatives modify properties of the $S_1... (\pi\pi^*)$ potential energy surface, including the nature of twisted intermediates and barriers to reach isomerising conical intersection seams[8,9,37–46]. This state of affairs highlights the need for unambiguous measurements of the gas-phase excited state dynamics on model PYP chromophores.

Here, photodetachment action, photoelectron and time-resolved photoelectron spectroscopy are used to probe the deprotonated methyl ester of para-coumaric acid, $p\text{CEs}^-$ (Fig. 1a). We uncover a decay mechanism that involves internal conversion from the bright $S_1(\pi\pi^*)$ valence-localised state to a DBS. Observation of an excited valence to non-valence state internal conversion in the intramolecular photophysics of an unsuspecting chromophore opens the questions about the commonness of such mechanisms in the excited state dynamics of isolated anions. The present results challenge the earlier understanding of this anion in the gas-phase in relation to a model PYP chromophore, by showing there is no internal conversion to the ground electronic state and therefore no concomitant $E \rightarrow Z$ isomerisation.

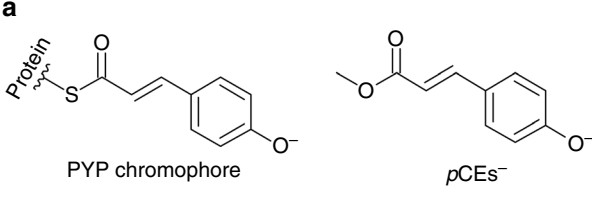

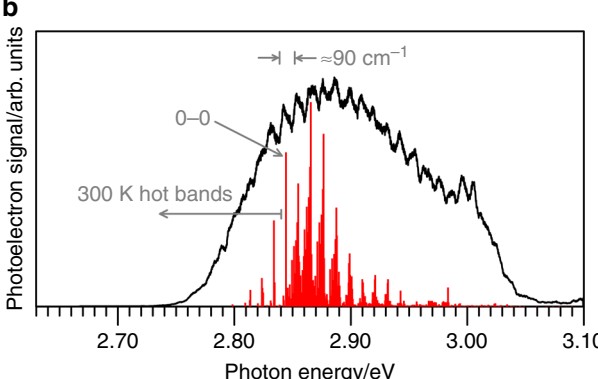

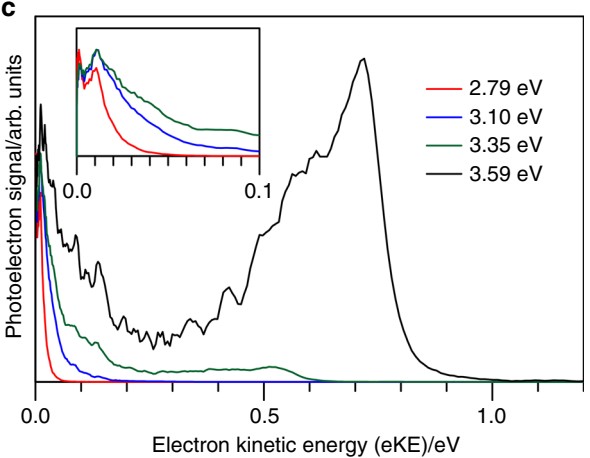

**Fig. 1 Single-colour action spectroscopy of the methyl ester of deprotonated para-coumaric acid ($p\text{CEs}^-$). a** Structure of the photoactive yellow protein (PYP) chromophore and the E-isomer of $p\text{CEs}^-$. **b** Photodetachment action spectrum of $p\text{CEs}^-$ (black) and Franck-Condon-Herzberg-Teller simulation of the $S_1(\pi\pi^*) \leftarrow S_0$ transition at 300 K (red). **c** Photoelectron spectra of $p\text{CEs}^-$ using $h\nu$ = 2.79 (red), 3.10 (blue), 3.35 (green), and 3.59 eV (black). The inset in **c** shows the low-eKE region.

## Results

**Photodetachment action spectroscopy.** Figure 1b shows a photodetachment action spectrum for $p$CEs$^-$ recorded over the $S_1(\pi\pi^*) \leftarrow S_0$ transition. The overall shape of the spectrum is in good agreement with earlier work[14,17], but additionally shows vibrational structure with $\approx 90$ cm$^{-1}$ spacing. A Franck-Condon-Herzberg-Teller simulation of the $S_1(\pi\pi^*) \leftarrow S_0$ absorption spectrum at 300 K, which has been translated to agree with an EOM-CC3/aug-cc-pVDZ vertical excitation energy of the $S_1(\pi\pi^*) \leftarrow S_0$ transition at 2.88 eV[17], is in good agreement with the experimental vertical excitation energy at $2.88 \pm 0.01$ eV. The simulation assigns the 0-0 transition at $2.85 \pm 0.01$ eV and predicts that the absorption spectrum is dominated by a progression involving an in-plane bending mode with a calculated wavenumber of $\nu_3 = 88$ cm$^{-1}$ (see Supplementary Note 1). The red-edge of the photodetachment spectrum is substantially broadened by hot-band transitions because the present experiments were conducted with ions thermalised to $\approx 300$ K.

**Single-colour photoelectron spectroscopy.** Four representative photoelectron spectra of $p$CEs$^-$ recorded using a tuneable light source ($\approx 6$ ns light pulse duration) are shown in Fig. 1c. Extrapolation of the high-electron kinetic energy (eKE) edge of the spectra provided the adiabatic detachment energy (ADE) at $2.83 \pm 0.05$ eV. The vertical detachment energy (VDE) is $2.87 \pm 0.02$ eV and refines the value given in an earlier photoelectron spectroscopy study[21]. The four photoelectron spectra have a low-eKE feature, i.e., eKE < 0.1 eV, see inset in Fig. 1c. In several earlier studies on $p$CEs$^-$ and other PYP chromophore derivatives[18–21], these low-eKE features were assigned to thermionic emission, which involves statistical electron ejection following recovery of the ground electronic state. However, our measurements do not agree with this assignment for several reasons. Firstly, the low-eKE peak has a clear double peak structure. Secondly, the photoelectron angular distribution, quantified with the $\beta_2$ anisotropy parameter[47], of the low-eKE feature have values ranging from $+0.1$ to $+0.2$, which implies that some fraction of electron detachment, leading to the low-eKE signal occurs on a timescale that is faster than molecular rotation (picoseconds). Thirdly, delaying the acquisition gate on the electron imaging detector by 50 ns relative to the light pulse showed that all photoelectron signal, including the low-eKE signal, appears on a <50 ns timescale following excitation. All of these observations contradict thermionic emission, which should have a statistical energy distribution, an isotropic angular distribution, and occur on the microseconds to milliseconds timescale. Instead, low-eKE features with the observed attributes can result from autodetachment of valence or non-valence states (picoseconds detachment timescale) that are energetically in close proximity to the detachment threshold, as is the case for $p$CEs$^-$. Elucidating the origin of these low-eKE electrons requires the excited state dynamics to be probed in real time.

**Time-resolved photoelectron spectroscopy.** The dynamics of the $S_1(\pi\pi^*)$ state in $p$CEs$^-$ were investigated using a femtosecond pump-probe strategy. A 2.83 eV pump photon ($\approx 0.02$ eV bandwidth) was resonant with red-edge of the action spectrum and a 1.55 eV delayed ($\Delta t$) probe photon monitored the evolution of the excited state population. Time-resolved photoelectron spectra, which show the transient changes in eKE with $\Delta t$, were determined by subtracting an average of the spectra with $\Delta t \leq -100$ fs (i.e., probe before pump) from all other spectra. Figure 2a shows a series of representative time-resolved photoelectron spectra over the first 3 ps of the dynamics. For three specific delays, we acquired higher single-to-noise spectra and these are shown in

Fig. 2b. The negative (bleach) signal for eKE < 0.2 eV has been truncated for clarity; the bleach signal comes about because some of the photoexcited $S_1(\pi\pi^*)$ population that leads to the low-eKE electrons is photodetached by the probe pulse, leading to a peak at higher kinetic energy. The $\Delta t = 0$ ps time-resolved spectrum shows that this feature has a spectral maximum at eKE $\approx 1.4$ eV. This feature subsequently broadens and red-shifts to a maximum at eKE $\approx 0.75$ eV with increasing $\Delta t$. The shift is exemplified by the spectra at $\Delta t = 2.5$ and 7.5 ps in Fig. 2b. However, the $\Delta t = 2.5$ ps time-resolved spectrum has an additional sharp feature at eKE $\approx 1.53$ eV that differs in appearance from $\Delta t = 0$ ps spectrum. The $\Delta t = 2.5$ ps velocity-map image (from which the time-resolved spectrum is derived) is shown as an inset in Fig. 2b. The photoelectron image shows that the detachment signal leading to the sharp feature is anisotropic, peaking along the probe laser polarisation axis ($\beta_2 \approx +1$), which suggests $p$-wave photodetachment induced by the probe[47]. A sharp pump-probe feature that peaks near the detachment photon energy with $p$-wave character is consistent with detachment from a non-valence state of the anion[27,48]. The sharp peak is evident in the $\Delta t = 0.3$–3.0 ps time-resolved spectra.

The integrated time-resolved photoelectron signal for eKE > 0.2 eV is shown in Fig. 2c and suggests at least two components: a fast decay on the sub-picosecond timescale and a slower decay over tens of picoseconds. The integrated signal fits well to a kinetic model in which an initial excited state decays with a lifetime of $\tau_1 = 0.6 \pm 0.1$ ps (fast evolution away from the initial broad high-eKE signal) into two intermediate states that separately decay with lifetimes $\tau_2 = 2.8 \pm 0.7$ ps (decay of the sharp-eKE feature at eKE $\approx 1.53$ eV) and $\tau_3 = 45 \pm 4$ ps (decay of the time-resolved signal peaking around eKE $\approx 0.75$ eV). The spectrum associated with lifetime $\tau_3$ is represented by the time-resolved spectrum at $\Delta t = 7.5$ ps in Fig. 2b, which decays with essentially no further change in its spectral distribution. The time-resolved depletion signal, i.e., eKE < 0.2 eV, returns to the zero level at long $\Delta t$. This confirms that all photoelectron signal contributing to the pump-only background spectrum (or single-colour spectra in Fig. 1c) has occurred on the sub-200 ps timescale. Unfortunately, no meaningful analysis could be performed on the depletion signal due to poor signal to noise (small difference signal on a large background), although the recovery kinetics appear to be consistent with the overall decay of $\tau_3$.

Further insight into the nuclear relaxation dynamics can be gained through the average kinetic energy, <eKE>, of the time-resolved signal (excluding signal for eKE < 0.2 eV) as a function of $\Delta t$ (Fig. 2d). The <eKE> data shows a rapid shift towards lower values with $\Delta t$, asymptoting to <eKE> = 0.75 eV after ~8 ps, consistent with the time-resolved spectrum at $\Delta t = 7.5$ ps (Fig. 2b). The <eKE> data fit well to a bi-exponential model with $\tau_1' \approx 0.6$ ps and $\tau_2' = 4.7 \pm 0.5$ ps (Fig. 2d). A single exponential with lifetime of $2.6 \pm 0.2$ ps provides a visibly poorer fit. The lifetimes $\tau_1'$ and $\tau_2'$ can be correlated with the lifetimes in the population dynamics from the kinetic model. Specifically, $\tau_1' \approx \tau_1$ and $\tau_2' \approx \tau_2 + 1.4$ ps, where 1.4 ps is the $\Delta t$ at which the sharp feature at eKE $\approx 1.53$ eV attributed to detachment from the non-valence state has reached its maximal contribution to the overall signal in the time-resolved spectra (Fig. 2c).

## Discussion

The time-resolved results for $p$CEs$^-$ can be interpreted according to the model presented in Fig. 3a–c. For $\Delta t \approx 0$, the broad, high-eKE time-resolved spectrum (Fig. 2a, b) is due to the probe pulse sampling the $S_1(\pi\pi^*)$ excited state in the Franck-Condon region of the anion (Fig. 3a). Over the ensuing $\approx 600$ fs ($\tau_1 = 0.6 \pm 0.1$ ps), the time-resolved eKE distribution shifts to lower <eKE> and

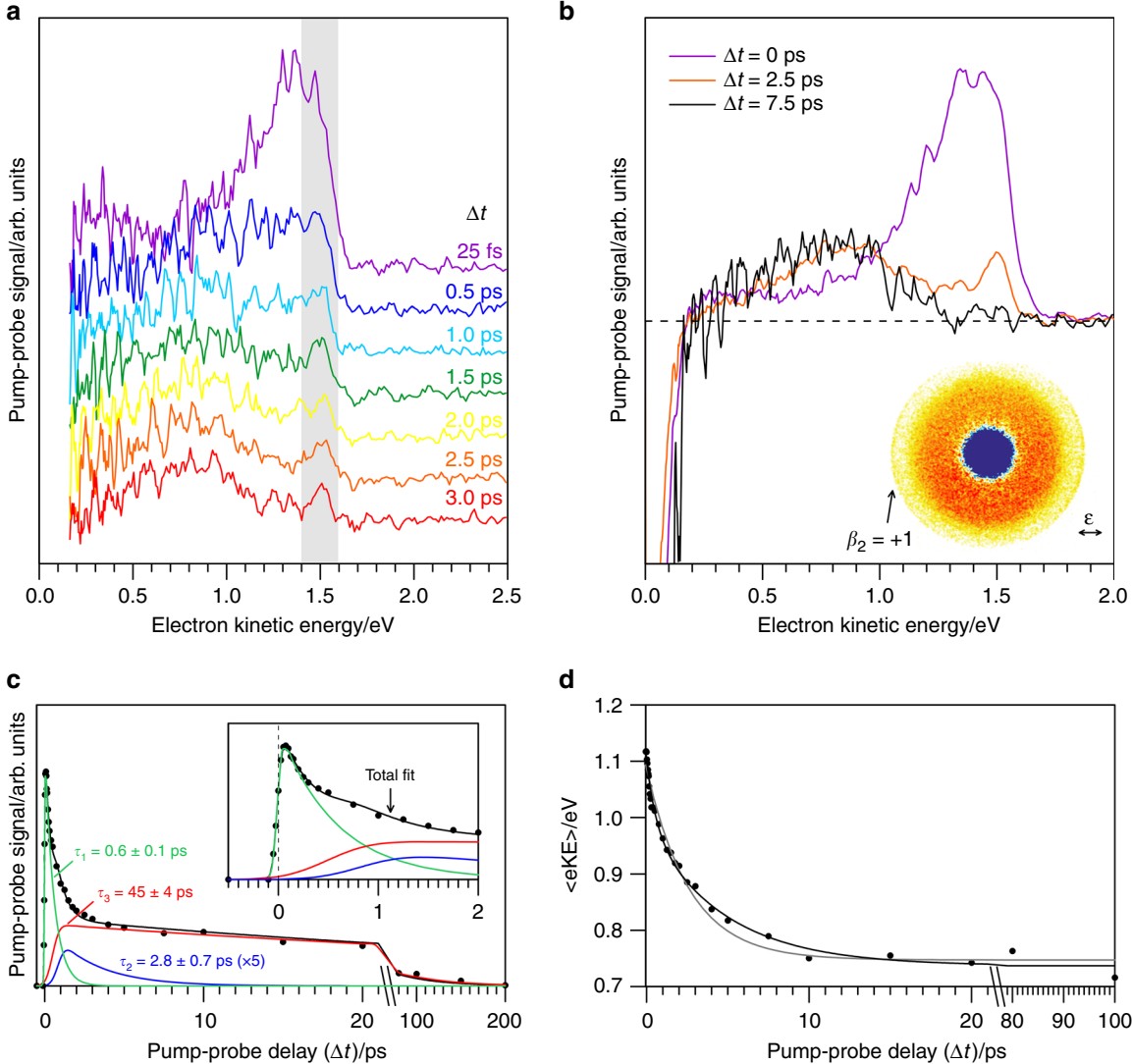

**Fig. 2 Time-resolved dynamics in $p$CEs⁻ following excitation of the $S_1(\pi\pi^*)$ state at 2.83 eV. a** Waterfall plot of the time-resolved spectra over the first 3 ps, in 0.5 ps increments. Grey shading highlights evolution of the dipole-bound state. **b** Example time-resolved spectra taken with higher signal-to-noise than **a**. The $\Delta t = 2.5$ ps spectrum illustrates the sharp, high-eKE feature associated with the DBS. The corresponding velocity-map is shown in the inset, with $\varepsilon$ indicating the femtosecond pump and probe pulse electric field polarisation. **c** Total time-resolved signal with $\Delta t$ and kinetic model fit, $\tau_1 = 0.6 \pm 0.1$ ps, $\tau_2 = 2.8 \pm 0.7$ ps, and $\tau_3 = 45 \pm 4$ ps. **d** Average time-resolved electron kinetic energy with $\Delta t$, which is best fit with two lifetimes (black trace), $\tau_1' \approx \tau_1$ and, $\tau_2' = 4.7 \pm 0.5$ ps $\approx \tau_2 + 1.4$ ps, with 1.4 ps being the $\Delta t$ for maximal DBS signal. The grey trace is a single-exponential fit, which is visually poorer.

signal from the non-valence state becomes evident, reaching maximum signal level at $\Delta t \approx 1.4$ ps (Fig. 3b). The time-resolved dynamics associated with the $\tau_1$ lifetime involve substantial changes in <eKE> (Fig. 2d), reflecting nuclear relaxation away from the Franck-Condon geometry. A non-valence state is populated along this trajectory, facilitated through a nonadiabatic transition. This state has a lifetime of $\tau_2 = 2.8 \pm 0.7$ ps and an average binding energy of eBE = $h\nu$ − eKE $\approx 20$ meV. Concomitant with appearance of the non-valence state is the appearance of time-resolved signal with a spectral maximum at < eKE > $\approx 0.75$ eV. The disappearance of the time-resolved feature associated with the non-valence state roughly agrees with $\tau_2' = 4.7 \pm 0.5$ ps $\approx \tau_2 + 1.4$ ps, indicating that the second time component in the <eKE> fit in Fig. 2d is strongly influenced by loss of the non-valence state population. Indeed, if the <eKE> data is analysed over the $0.2 <$ eKE $< 1.3$ eV window, i.e., excluding the signal associated with the non-valence state, the integral can be well fit with a single exponential having a lifetime within the uncertainty of $\tau_1' \approx \tau_1 = 0.6 \pm 0.1$ ps.

The fraction of excited state population that remains in the $S_1(\pi\pi^*)$ state, which is shown by the $\Delta t = 7.5$ ps spectrum in Fig. 2b, decays with a lifetime of $\tau_3 = 45 \pm 4$ ps. The broad spectral shape and the maximum eKE of this time-resolved feature indicates a valence state with electron-binding energy, eBE = $1.55 - 0.75$ eV $\approx 0.8$ eV, consistent with detachment from the valence-localised $S_1(\pi\pi^*)$ state along some nuclear displacement away from the Franck-Condon region. The relatively long lifetime of this time-resolved signal combined with the absence of spectral evolution suggests that the $S_1(\pi\pi^*)$ population becomes trapped in a potential energy minimum that is presumably associated with a twisted intermediate (phenyl group and alkene tail twisted by 90°) as predicted by theory[8,9]. In principle, this signal could also arise from the lowest triplet state of the anion. However, the triplet state has not been invoked in previous experimental or computational work and would not be consistent with the observed photoisomerisation of the chromophore in PYP or coumaric acid anions in solution. The twisted intermediate decays through vibrational autodetachment since the time-resolved

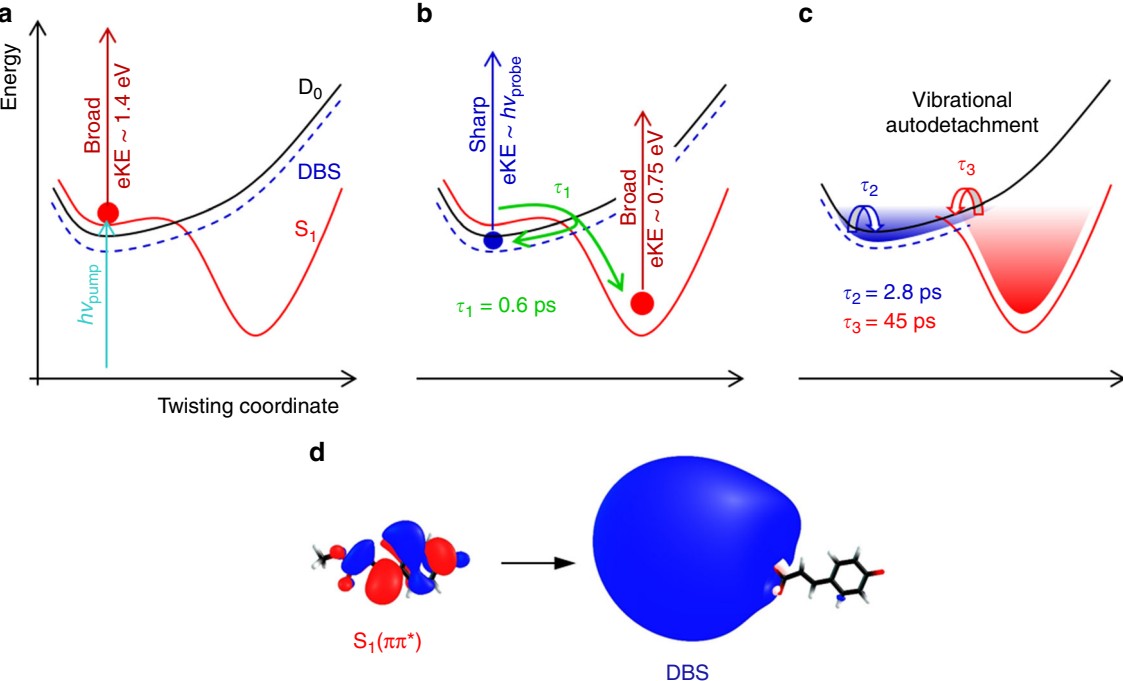

**Fig. 3 Interpretation of the time-resolved dynamics for $p$CEs$^-$. a** At $\Delta t \approx 0$, $S_1(\pi\pi^*)$ population leads to a broad, high-eKE distribution. **b** Rapid nuclear motion away from the Franck-Condon geometry to a twisted intermediate ($\tau_1 = 0.6 \pm 0.1$ ps) gives rise to the broad signal peaking around eKE $\approx 0.75$ eV. Concurrently, there is internal conversion to a DBS, leading to the sharp eKE signal at eKE $\approx h\nu_{probe}$. **c** DBS and twisted intermediate populations decay through vibrational autodetachment with lifetimes $\tau_2 = 2.8 \pm 0.7$ and $\tau_3 = 45 \pm 4$ ps, respectively. **d** CCSD natural orbitals for the $S_1(\pi\pi^*)$ state and DBS at the initial anion geometry (80% isodensity surface).

measurements show recovery of the depletion signal and there was no evidence for thermionic emission in the single-colour spectra. Our calculations on the twisted intermediate predict that it lies vertically $\approx 0.8$ eV below the detachment threshold, in excellent agreement with the experimental data.

The time-resolved signal associated with non-valence state in $p$CEs$^-$ is assigned to a DBS since the neutral core of $p$CEs has $\mu = 3.2$ or $3.7$ D at the anion or neutral equilibrium geometries, respectively, which satisfies the $\mu > 2.5$ D condition to support a DBS[23,24]. Excited state calculations as part of the current work characterised a DBS with binding energy of 4 and 8 meV at the anion and neutral equilibrium geometries, respectively. Although these binding energies are lower than the average experimental value of $\approx 20$ meV, $p$CEs$^-$ was initially thermalised to 300 K and will exist in several conformations with each possessing a different dipole moment of the neutral core. The orbital associated with the DBS at the anion equilibrium geometry is shown in Fig. 3d.

The model used to fit the integrated time-resolved signal for $p$CEs$^-$ in Fig. 2c assumes that the DBS decays through autodetachment with a lifetime of $\tau_2 = 2.8 \pm 0.7$ ps, i.e., the DBS population does not convert back to the valence-localised state. Further evidence for autodetachment from the DBS can be seen in the single-colour spectra (Fig. 1c), which show that the low-eKE signal has a double peak structure. Structured, low-eKE features in photoelectron spectra have been previously observed in the vibrational autodetachment dynamics from non-valence states in cluster systems that were initially thermalised to room temperature[27,48], with the spectral structure attributed to specific vibrational modes that lead to a substantial change in dipole moment of the neutral core and "shake off" the loosely bound non-valence electron[49]. The timescale for vibrational autodetachment from the DBS in the earlier study was $\approx 2$ ps, which is in line with the $\tau_2 = 2.8 \pm 0.7$ ps lifetime observed for $p$CEs$^-$. The

time-resolved data show that the DBS feature has disappeared after several picoseconds, for example see the $\Delta t = 7.5$ ps time-resolved spectrum in Fig. 2b, but the spectra still show photo-depletion signal in the low-eKE region. Since the photodepletion signal recovers to zero at long $\Delta t$, we conclude that excited state population in the twisted intermediate geometry also leads to low-eKE electrons through vibrational autodetachment. In principle, vibrational autodetachment from the twisted intermediate could be mode-specific, leading to a structured low-eKE feature, although the time-resolved data associated with the low-eKE photodepletion signal was too noisy to determine the origin of the low-eKE structure in the single-colour spectra (Fig. 1c).

It is worth noting that even though the time-resolved images and spectra clearly show formation of a DBS, we do not know the branching ratio associated with formation of the DBS and the nuclear dynamics leading to the twisted intermediate because the relative photodetachment cross sections (with probe pulse) are not known. Intuitively we expect the photodetachment cross section for the DBS to be large. Johnson and coworkers[50] proposed that the photodetachment cross section depends on the detachment wavelength and maximises when the de Broglie wavelength ($\lambda_{dB}$) of the outgoing photoelectron is similar to spatial extent of the DBS orbital. At the probe energy of 1.55 eV, $\lambda_{dB} = 9.9$ Å, which is comparable to the radius of the 80% iso-density surface at $\sim 12$ Å for the DBS orbital shown in Fig. 3d.

Internal conversion between the $S_1(\pi\pi^*)$ state and a DBS requires vibrational motion that induces a nonadiabatic coupling between the two states. The prototypical example, albeit for the ground valence state, is the nitromethane anion in which the nitro group is planar for the DBS but tilted for the valence ground state[51]. This tilting distortion induces a curve crossing between the two states[52]. The reverse process, internal conversion from a DBS to a valence state, has been observed in several cluster systems, including I$^-\cdot$X clusters, where X has included $(H_2O)_n$ or

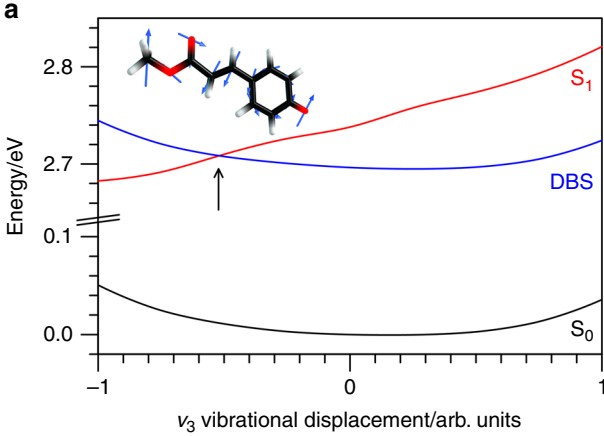

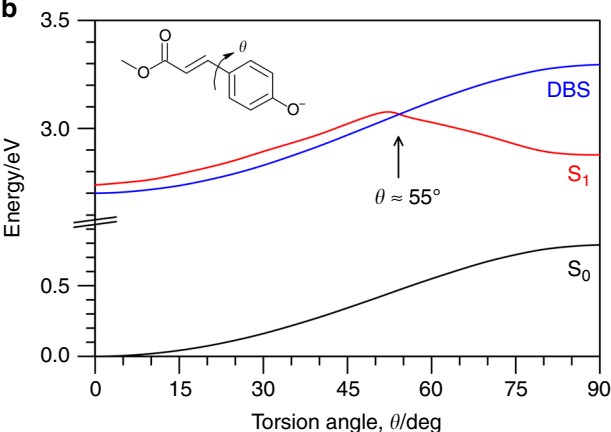

**Fig. 4 Potential energy surfaces for the $S_1(\pi\pi^\star)$ valence-localised state and the dipole-bound state.** The potential energy surfaces are represented by a black line for the ground state, a red line for the $S_1(\pi\pi^\star)$ valence-localised state, and a blue line for the dipole-bound state (DBS), respectively. **a** Potential energy surface calculation for motion along the principal Franck-Condon active vibration, $\nu_3$, with displacement vectors shown on the molecular structure in the inset. 0 corresponds to the initial anion geometry, −1 and 1 correspond to turning points for the ground vibrational level. **b** Potential energy surface calculation for rigid torsion scan about the single-bond connecting the phenyl and alkene tail by angle $\theta$, as shown inset, with 0° corresponding to the initial anion geometry and 55° to a crossing of valence and non-valence surfaces. Note, the calculated $S_1(\pi\pi^\star)$ vertical excitation energy and VDE (not shown) at the anion equilibrium geometry are both ≈ 0.1 eV lower than experiment.

small organic species[53–56]. In the case of I⁻·nucleobase clusters, theory suggested that ring distortion vibrations were responsible for the non-valence-to-valence coupling[57]. Similar stretching and wagging vibrations were proposed to explain to the valence-to-non-valence internal conversion dynamics characterised in an anionic quinone cluster system[27]. In a similar vein to these earlier studies, two possible coupling pathways between the $S_1(\pi\pi^\star)$ state and the DBS in $p$CEs⁻ determined from potential energy surface calculations are shown in Fig. 4. The first pathway (Fig. 4a) is associated with motion along the principal Franck-Condon active vibrational mode, $\nu_3$, and indicates a crossing between the $S_1(\pi\pi^\star)$ state and DBS potential energy surfaces roughly halfway along the displacement coordinate in one direction. A second possible pathway (Fig. 4b) involves torsion of the single-bond connecting the phenoxide to the alkene-ester tail, which was identified as principal nuclear displacement required to reach the twisted intermediate from excited state molecular dynamics simulations

on the related thioester chromophore of PYP[8,9]. A rigid potential energy surface scan about dihedral angle $\theta$, suggests that at $\theta \approx 52°$ there is a barrier on the potential energy surface for the $S_1(\pi\pi^\star)$ valence-localised state of ≈0.4 eV (likely lower in a relaxed potential energy surface scan), and a curve crossing with the DBS at $\theta \approx 55°$. Presumably, there are further coupling pathways along different nuclear displacement coordinates. Excited state molecular dynamics simulations with provision to describe the DBS are needed to identify the active coupling pathways.

The excited state dynamics for $p$CEs⁻ can be compared with those measured for the deprotonated methyl ketone derivative excited at 3.10 eV (400 nm)[22]. The interpretation of those time-resolved dynamics was that most of the photoexcited population evolved to a twisted intermediate, which then internally converts to the ground electronic state with some fraction undergoing isomerisation by passage through a conical intersection[58]. However, because the deprotonated methyl ketone study utilised a magnetic bottle analyser, the experiment was unable to reliably accumulate very low-eKE signal and, hence, provide conclusive evidence for the dynamics associated with that signal. The excited state lifetimes were 1 and 52 ps, which are comparable to those for $p$CEs⁻ ($\tau_1 = 0.6 \pm 0.1$ ps and $\tau_3 = 45 \pm 4$ ps), and were similarly assigned to nuclear relaxation away from the Franck-Condon geometry to form a twisted intermediate and its subsequent decay[8,9]. For $p$CEs⁻, the lack of ground state recovery, implies there is no $E \rightarrow Z$ photoisomerisation by passage through a conical intersection. This conclusion is consistent with recent ion mobility mass spectrometry measurements, which found no evidence for $E \rightarrow Z$ isomerisation for photon energies resonant with the $S_1(\pi\pi^\star) \leftarrow S_0$ transition[17]. The striking disparity in ground state recovery dynamics between $p$CEs⁻ and the methyl ketone derivative is presumably linked with substantial differences in potential energy surface topologies associated with isomerising conical intersections[8,40,42] for which the isolated chromophores present ideal theoretically tractable benchmarks.

In the context of photoactive yellow protein, the lack of ground state recovery dynamics for isolated $p$CEs⁻ differs from that for the thioester chromophore in the protein, whereby ground state recovery with concomitant $E \rightarrow Z$ isomerisation is the key actinic step in the protein's photocycle. Assuming $p$CEs⁻ behaves similarly to the isolated thioester derivative, the difference between the photo-induced dynamics in vacuo and in the protein implies that protein side group interactions within the chromophore-binding pocket are crucial for facilitating an ultrafast photo-isomerisation of the chromophore.

Finally, we briefly consider the DBS of the chromophore anion in the context of the PYP protein. As the DBS is a weakly bound state, it only exists over a narrow energy range just below the detachment threshold. In the condensed phase, the detachment energy of the chromophore increases due to stabilisation of the ground anion state, shifting the DBS energy away from the energy for the $S_1(\pi\pi^\star)$ state and possibly closer to higher-lying valence-localised states (e.g., $S_2$ or $S_3$). The question of whether a DBS exists in a condensed phase environment is contentious and is presumably linked with local density around the chromophore. In a dense environment, the excluded volume would inhibit the existence of a DBS. On the other hand, charge-transfer-to-solvent states that exist near the detachment threshold of several anions in solution can be viewed as a DBS supported by the dipole of the solvent[59]. In the PYP protein, the chromophore exists in a binding pocket, which has a lower local density compared with solution and could conceivably support a DBS. For example, recent experiments have shown that a DBS can survive in close proximity to molecular fragments[60].

There are two key conclusions of the work. Firstly, we have demonstrated the existence of a valence-to-non-valence state

internal conversion process in the intramolecular photophysics of a common chromophore anion, which is active on the picosecond timescale. The formation of a dipole-bound state through internal conversion from an excited valence-localised state for an unsuspecting chromophore raises questions into how common this process might be, and also highlights that care must be taken when assigning low-eKE photoelectron features to dynamical processes implied from single-colour photoelectron spectra. Many biochromophore anions possess $S_1(\pi\pi^\star)$ states that extend into the detachment continuum and have large dipole moments for the neutral state. For example, the methyl ketone derivative of $p$CEs$^-$ has $\mu = 3.4$ D for the neutral core of the $E$-isomer at the anion geometry, which is similar to that for $p$CEs ($\mu = 3.2$ D), however, there was no evidence for DBS formation in the time-resolved spectra[22]. Similar situations exist for $p$HBDI$^-$ ($\mu = 7.5$ D for neutral $Z$-$p$HBDI) and deprotonated retinoic acid ($\mu = 18.2$ D for neutral all-$trans$ isomer), which are the chromophore in green fluorescent protein and a retinoid molecule, and have action spectra that extend over their detachment thresholds[12,61,62]. There is no experimental evidence for coupling with non-valence states in these anions.

A second key conclusion is that $p$CEs$^-$ does not isomerise following photoexcitation of the $S_1(\pi\pi^\star)$ state, despite the anion's structural similarity to the chromophore in photoactive yellow protein. Instead, the excited state decays by vibrational auto-detachment. The signature of this in photoelectron spectroscopy is low-energy electron emission, although the corresponding features in photoelectron spectra can be difficult to distinguish from signal associated with ground state thermionic emission. The application of time-resolved photoelectron spectroscopy removes the interpretation ambiguity and provides a direct measurement of the excited state dynamics. The dramatically different dynamics for small chemical changes in the chemical composition of PYP model chromophores indicates the extreme sensitivity of the $S_1(\pi\pi^\star)$ potential energy surface to functional group modification, and means that PYP chromophores are ideal benchmarks for theoretically tractable excited state dynamics simulations in anions.

## Methods

**Experimental**. Experiments were performed using a custom-built photoelectron imaging apparatus[63–65]. Gas-phase $p$CEs$^-$ was produced through electrospray ($-5$ kV) of a ~1 mM methanolic solution of the sample (99%, Sigma-Aldrich, shielded from light) with a trace of ammonia to aid deprotonation. Electrosprayed anions were transferred via a vacuum transfer capillary into a radio frequency ring-electrode ion trap. The trapped anions were unloaded (10 or 50 Hz) into a colinear time-of-flight optics assembly that accelerated the ions along a 1.3 m flight region toward a continuous-mode penetrating field velocity-mapping assembly[66]. Laser pulses were timed to interact with the mass-selected ion packet at the centre of the velocity-map imaging stack. Ejected electrons were velocity mapped onto a dual (chevron) multi-channel plate detector, followed by a P43 phosphor screen, which was monitored with a charge-coupled device camera. Velocity-map images were accumulated with a 500 ns multi-channel plate acquisition gate; delay of the gate relative to the laser pulse by 50 ns demonstrated no thermionic emission signal[67]. Velocity-mapping resolution was ~5% and the electron kinetic energy (eKE) scale was calibrated from the spectrum of I$^-$. Velocity-map images were processed using an antialiasing and polar onion-peeling algorithm[68], providing the photoelectron spectra and associated angular distributions in terms of $\beta_2$ values[47].

The photodetachment action spectrum and single-colour photoelectron spectra were recorded using light from a Continuum Horizon optical parametric oscillator pumped with a Continuum Surelite II Nd:YAG laser (~0.5 mJ pulse$^{-1}$, unfocused).

Femtosecond laser pulses were derived from a Spectra-Physics Ti:sapphire oscillator and regenerative amplifier. The 2.83 eV (438 ± 5 nm, ~20 μJ) pump pulses were produced by fourth-harmonic generation (two successive BBO crystals) of the idler output from an optical parametric amplifier (Light Conversion TOPAS-C). The 1.55 eV (800 nm, ~100 μJ) probe pulse is the fundamental of the laser. Pump and probe pulses were delayed relative to each other ($\Delta t$) using a motorised delay line. Both pulses were combined collinearly using a dichroic mirror and were loosely focused into the interaction region using a curved metal mirror. The pump-probe cross correlation was ~70 fs.

**Theoretical**. Franck-Condon-Herzberg-Teller simulations were performed at 300 K at the ωB97X-D/aug-cc-pVDZ level of theory within the TD-DFT framework using Gaussian 16.B01[69–72]. DBS calculations were performed at the EA-EOM-CCSD/GEN (ROHF reference) level of theory[73] using CFOUR 2.1[74]. Potential energy surface calculations of the $S_1(\pi\pi^\star)$ state and VDE were performed at the STEOM-DLPNO-CCSD/aug-cc-pVDZ level of theory using ORCA 4.2.0[75]. Molecular basis set GEN is the $6–31+G^*$ basis set[76] with eight additional $s$, $p$, and $d$ diffuse functions centred 1 Å beyond the CH$_3$ on the ester functional group, i.e., at the positive end of the electric dipole moment associated with the neutral core. Orbital exponents for the additional diffuse functions followed the recommendations in Skurski et al.[77], starting at 0.1 and followed a decreasing geometric progression with factor 3.2. Calculations in which the additional diffuse functions were situated 2 or 3 Å beyond CH$_3$ group or 1 Å between the CH$_3$ group and carbonyl oxygen atom gave no substantial change to the DBS-binding energies. Neither further diffuse functions nor change of the geometric progression factor to 6.4 gave an increased DBS-binding energy.

## Data availability
The data that support the findings of this study are available from the authors upon request.

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

## Acknowledgements

This research was supported by a University of East Anglia start-up allowance (JNB), the European Research Council, grant 306536 (CAS, JRRV) and the EPSRC grant EP/D073472/1 (JRRV).

## Author contributions

J.R.R.V. and J.N.B. designed and conceived the experiments. J.N.B. and C.S.A. conducted the experiments. J.N.B. conducted calculations and the analysis of the dynamics. J.N.B. and J.R.R.V. wrote the manuscript.

## Competing interests

The authors declare no competing interests.
