## [Peer Review File · Nature Communications]

Reviewers' comments:

Reviewer #1 (Remarks to the Author):

The present manuscript reports an experimental study using femtosecond time-resolved photoelectron spectroscopy to follow the excited-state dynamics from the initially excited valence (π,π^*) state of an anionic chromophore found in the photoactive yellow protein. This work is complemented by theoretical calculations. The main result is that an internal conversion between the initially excited valence (π,π^*) state and a dipole-bound (Rydberg) state is observed on an ultrafast timescale. This poses the question whether this phenomenon is rather common in anions, despite the fact it has rarely been reported. A second conclusion is that there is no trans-cis photoisomerization of the studied chromophore in the gas phase.

In my opinion, this study adds significant new insight into the mechanistic picture of the excited-state behaviour of this type of chromophore. The conclusion of the manuscript is obviously of very broad impact for photophysicists and photochemists. In addition, the manuscript is well written and is concise. Thus I believe that this manuscript can be published in Nature Communications with only very minor changes suggested below.

1) p. 4, line 77: two significant articles could be added to the list of references on the effect of the environment (solvent and protein) on the excited-state potential energy surface and access to the S₀/S₁ conical intersection in PYP chromophores:

Boggio-Pasqua et al., J. Am. Chem. Soc. 2009, 131, 13580-13581.

Boggio-Pasqua et al., Phys. Chem. Chem. Phys. 2012, 14, 7912-7928.

2) There is an inconsistency about the value of the tau₃ lifetime across the manuscript. It is written 45 ps on pages 8, 9, 11 and 16, but the value reported in the caption of Figure 3 and also in Figure 3c on p. 12 is 54 ps.

3) p.14, line 295: the S₁ excited-state barrier of ca. 0.4 eV is estimated from the potential energy profile in Figure 4b along a scan of the torsion angle around the C=C double bond. But it is not explained how this can be performed (rigid or relaxed scan?). I believe the S₁ potential energy profile results simply from a rigid scan in which case the maximum along the S₁ potential energy profile is a very poor approximation of the transition state. The barrier is thus largely overestimated in such a case. I think this should be mentioned. A better way to proceed would have been to use the geometries resulting from a minimum energy path calculation on S₁ using for example CASSCF, and then run the EOM-CCSD single point calculations at these geometries.

4) p. 15: There is an inconsistency in the colour code in Figure 4. The S1(π,π^*) potential energy curve is in red, while that of the dipole-bound state is in blue. But the colour of the labels S1 and DBS are inverted (S1 in blue font and DBS in red font).

5) p. 19: It should be written that the FCHT spectra were simulated using time-dependent density functional theory (TD-DFT).

Typos:

p. 8, line 149 : correct 'The image and shows...'

p. 12, figure 3 caption: correct '...twisted intermediate population populations decay through...'

Reviewer #2 (Remarks to the Author):

The submitted manuscript entitled “**Ultrafast valence to non-valence excited state dynamics in a common anionic chromophore**” by James N. Bull *et al.* discusses new experimental findings regarding the excited state dynamics in the deprotonated methyl ester of para-coumaric acid (pCEs⁻), a model chromophore for the photoactive yellow protein.

The manuscript consists of two main experiments, with the resulting conclusions supported by excited state calculations. In a first one-color photoelectron spectroscopy experiment, Bull and co-workers find a fast, bi-modal and anisotropic low electron kinetic energy distribution upon excitation at different photon energies, which hints to a process different to the commonly accepted thermionic emission dynamics in such systems.

Further inspection of the dynamics on the excited state is performed using a femtosecond pump-probe method, from which time-dependent photoelectron spectra can be extracted. The authors find three distinct photoelectron distributions at different pump-probe delays, and explain the bi-modal, anisotropic and fast low electron kinetic energy component of the initial photoelectron spectra as arising from two different vibrational autodetachment processes. Some of the excited anions get trapped in a twisted intermediate on the S1 excited state and decays with a time-decay of about 45ps, while a second faster process involves formation of a non-valence dipole-bound state of the anion through valence to non-valence internal conversion dynamics (time decay of 2.8ps). The experimental demonstration of the latter intramolecular process is well supported by adequate modelling of the excited state potential energy surface and its crossing with the dipole bound surface and represents an unprecedented fundamental finding in an anionic biochromophore. Furthermore, the results presented here show that gas phase pCEs⁻ does not photo-isomerize, in contrast to the very similar chromophore present in photoactive yellow protein. This finding implies that subtle structural changes can drastically affect the excited state dynamics in very similar molecules and sets the stage for in-depth computational efforts and future careful analysis of time-independent photoelectron spectra.

This manuscript provides fundamental insight into an internal conversion process not observed to date and potentially affecting many anionic biochromophores. The finesse and detail of the experimental procedure, as well as the adequate supporting computational modelling, provides an unambiguous demonstration of this novel excited state intramolecular feature. The employed state-of-the-art methodology is unique and shows to be crucial to disentangle otherwise ambiguous photoelectron spectroscopy results. In addition, the writing style of the manuscript is excellent, in line with most of the manuscripts coming from this research group. The way the context and the results are presented makes it very easy to follow and comprehend the complex content of this study. In particular, the quality of Figure 3 is remarkable, as it discusses in a detailed but visually appealing way all the dynamical features occurring to pCEs⁻ following photoexcitation.

In my view, this study provides unexpected and broadly appealing fundamental insight into the excited state dynamics of model ionic biochromophores- molecular species of broad interest in fields ranging from biomedicine to molecular physics- at an unprecedented level of detail. I consider that the topic, quality and level of presentation of this manuscript is up to the high standards of *Nature Communications* and strongly recommend publication in this journal.

I ask the authors to address following minor points (further review is not needed):

- p2, l24: please insert a “comma” between “might be” and “given” to make the sentence more clear
- p7, l127: “following” instead of “flowing”
- p8, l149: please correct sentence: “The image and shows that the detachment...”

- p13, l257: please remove the “e.g”, as one can actually only see the disappearance at the 7.5 ps spectrum, not at several spectra
- Finally, I find the chosen spectra shown in Fig 2a to be slightly disturbing if compared with the numbers in the discussion. For instance, the spectrum at $\Delta(t)=2.5$ ps is shown as representative of the sharp peak arising from the DBS. However, when discussing this channel the authors always mention the 1.4 ps (p10, l198), and they claim a lifetime of 2.8ps, very close to the 2.5 ps in the figure. While probably the spectra with best signal to noise were chosen, and also the ones where signals from the three different channels do not overlap, it would have been perhaps better to choose numbers more according to the numbers discussed later.
- Perhaps it would be good to mark, in Figure 4, the 0.4 eV barrier that the authors mention in the manuscript along the torsion angle on the S1 potential energy surface, as well as the D0 curve.

Reviewer #3 (Remarks to the Author):

The manuscript presents the ultrafast dynamics of the deprotonated para-coumaric methyl ester, a model for the important chromophore in photoactive yellow protein. There are two important observations made in this manuscript: 1) The role of non-valence dipole bound state in the dynamic 2) The absence of an E to Z isomerization following the excitation, unlike the actual chromophore in PYP. The manuscript is concisely and clearly written and the conclusions are well supported. This manuscript is suitable for Nature Chem after consideration of the following points:

- 1) The data in Figure 2b and 2c should have error bars. The analysis hinges on fitting subtle features of these curves and large error bars would invalidate some of it. The PE image is very useful in that regard, because the dipole-bound feature shows up clearly. Showing a waterfall plot of the TR-PES or curves representing the time-evolution of the PEs signal within a specific eKE range could better highlight the dynamics.
- 2) It would be useful to have a bit more discussion on the overall electronic states of pCEs-. Is there any state, other than S1 and the dipole-bound state, which are expected nearby and could provide an alternative explanation of the observed TR-PES? For example, I would expect a triplet state, at lower energy than the S1 that could potentially explain the broad feature at $eKE=0.75$ eV.
- 3) The dipole-bound state observed here is, I think, purely a gas-phase phenomenon. It would be nice to have a bit of discussion on the implication for a solvated chromophore. Would the equivalent state be a charge transfer to solvent or nearby electron acceptor?

Reviewer #1:

1) p. 4, line 77: two significant articles could be added to the list of references on the effect of the environment (solvent and protein) on the excited-state potential energy surface and access to the S₀/S₁ conical intersection in PYP chromophores:

Boggio-Pasqua et al., J. Am. Chem. Soc. 2009, 131, 13580-13581.

Boggio-Pasqua et al., Phys. Chem. Chem. Phys. 2012, 14, 7912-7928.

We have now added these references.

2) There is an inconsistency about the value of the tau₃ lifetime across the manuscript. It is written 45 ps on pages 8, 9, 11 and 16, but the value reported in the caption of Figure 3 and also in Figure 3c on p. 12 is 54 ps.

This is a great spot and an oversight on our part. The number should be that given in the text and we have made the relevant change in the caption and figure.

3) p.14, line 295: the S₁ excited-state barrier of ca. 0.4 eV is estimated from the potential energy profile in Figure 4b along a scan of the torsion angle around the C=C double bond. But it is not explained how this can be performed (rigid or relaxed scan?). I believe the S₁ potential energy profile results simply from a rigid scan in which case the maximum along the S₁ potential energy profile is a very poor approximation of the transition state. The barrier is thus largely overestimated in such a case. I think this should be mentioned. A better way to proceed would have been to use the geometries resulting from a minimum energy path calculation on S₁ using for example CASSCF, and then run the EOM-CCSD single point calculations at these geometries.

The reviewer is indeed correct. We have performed a rigid scan about the C–C bond indicated by the torsion angle in Figure 4b. We agree that a relaxed scan would be desirable, however, we do not feel that such a calculation would add much to the present study. The main purpose of the calculations was to highlight that there are clear pathways that lead to strong coupling between the valence and non-valence states in the anion, rather than to definitively identify a single coordinate. Note that, in the manuscript, we do not claim that any of these are definitive and we point out that other pathways may indeed be possible.

We have updated the relevant text in the manuscript to clarify that our calculation was a rigid scan and that the barrier height was likely overestimated.

4) p. 15: There is an inconsistency in the colour code in Figure 4. The S₁(π,π*) potential energy curve is in red, while that of the dipole-bound state is in blue. But the colour of the labels S₁ and DBS are inverted (S₁ in blue font and DBS in red font).

We have updated the figure to have the correct colour label.

5) p. 19: It should be written that the FCHT spectra were simulated using time-dependent density functional theory (TD-DFT).

Agreed. We have amended the methods section.

Typos:

p. 8, line 149 : correct 'The image and shows...'

p. 12, figure 3 caption: correct '...twisted intermediate population populations decay through...'

Typos have been corrected.

Reviewer #2

- p2, l24: please insert a "comma" between "might be" and "given" to make the sentence more clear
- p7, l127: "following" instead of "flowing"
- p8, l149: please correct sentence: "The image and shows that the detachment..."
- p13, l257: please remove the "e.g", as one can actually only see the disappearance at the 7.5 ps spectrum, not at several spectra

These typos have been corrected.

- Finally, I find the chosen spectra shown in Fig 2a to be slightly disturbing if compared with the numbers in the discussion. For instance, the spectrum at $\Delta(t)=2.5\text{ps}$ is shown as representative of the sharp peak arising from the DBS. However, when discussing this channel the authors always mention the 1.4 ps (p10, l198), and they claim a lifetime of 2.8ps, very close to the 2.5 ps in the figure. While probably the spectra with best signal to noise were chosen, and also the ones where signals from the three different channels do not overlap, it would have been perhaps better to choose numbers more according to the numbers discussed later.

This is a fair point by the reviewer and they are (partially) correct. The 2.5 ps spectrum was chosen for its best clarity in identifying the narrow spectral feature for the dipole-bound state. At earlier pump-probe delay, the contribution from the valence state at this energy obscures the non-valence spectral feature. However, it is still of course there (and can be seen in angular distributions). The spectra displayed in the figure were actually acquired to achieve higher signal to noise to more emphasise the point. In response to reviewer 3, we have modified Figure 2 to include a waterfall plot of the dynamics over the first 3 ps in steps of 0.5 ps, which we feel shows why we have chosen the specific spectrum and that the DBS signature is present in all other spectra.

- Perhaps it would be good to mark, in Figure 4, the 0.4 eV barrier that the authors mention in the manuscript along the torsion angle on the S1 potential energy surface, as well as the D0 curve.

This is a rigid scan (see response to Reviewer 1) such that the barrier is likely to be an inaccurate and overestimated. Hence, there is little value in highlighting it in the figure and the barrier.

With regards to including D_0 , we understand the reviewer's point and actually tried to include this. The problem is that – on the scale shown in the figure – the D_0 and DBS surfaces are indistinguishable because the DBS is bound by <10 meV. So to avoid confusion, we have simply not included it and would prefer to keep it this way.

Reviewer #3 (Remarks to the Author):

1) The data in Figure 2b and 2c should have error bars. The analysis hinges on fitting subtle features of these curves and large error bars would invalidate some of it. The PE image is very useful in that regard, because the dipole-bound feature shows up clearly. Showing a waterfall plot of the TR-PES or curves representing the time-evolution of the PEs signal within a specific eKE range could better highlight the dynamics.

The reviewer's makes a fair remark, however, true error bars are in fact very difficult to obtain in these types of experiments and, indeed, most similar experiments do not have error bars for the same reason. Moreover, while the analysis relies on these curves, the spectra clearly show that the proposed dynamics are occurring. To emphasise this, we have taken the reviewers suggesting to show a waterfall plot of the data so that the reader can have a sense of the signal levels. This clearly shows the evolution of the signals. We refrain from plotting curves over specific eKE ranges because the features are so strongly overlapping. Overall, we feel that our data concerning the early-time dynamics (wavepacket bifurcation) only has the interpretation offered in the paper.

2) It would be useful to have a bit more discussion on the overall electronic states of pCEs-. Is there any states, other than S₁ and the dipole-bound state, which are expected nearby and could provide alternative explanation of the observed TR-PES? For example, I would expect a triplet state, at lower energy than the S₁ that could potentially explain the broad feature at eKE=0.75eV.

Again, the reviewer makes a fair point. In previous work on this and related molecular anions, only the S₁ state has ever been considered. Nevertheless, there should of course be a triplet state; the S₂ is significantly higher in energy and does not contribute. We have calculated the triplet energy and the photoelectron signal from this should peak at 0.8 eV, which is close to that expected from the twisted intermediate on the S₁ state. We, however, do not consider this to be a likely pathway because intersystem crossing has never been observed in any experiments (it would be clear in experiments using optical probes – e.g. transient absorption) and has not been identified previously in the many theoretical studies on this and related systems. Moreover, if the triplet state was involved, then PYP (and the pCK chromophore studied by Zewail) would not be able to isomerise through the observed pathway. We have now included a short mention of the triplet state in the manuscript.

3) The dipole-bound state observed here is, I think, purely a gas-phase phenomena. It would be nice to have a bit of discussion on the implication for a solvated chromophore. Would the equivalent state be a charge transfer to solvent or nearby electron acceptor?

This is a very interesting point that is still up for much debate. The best-known example of such a state is the charge-transfer-to-solvent states that can be viewed as dipole-bound states supported by the solvent. In general, however, solvating the anion will increase the detachment energy quite significantly and so we do not foresee that the DBS near S₁ will play any role in solution. However, it may for higher-lying excited states. In proteins, the stabilisation is lower and the dielectric constant is much lower than for water for example. We have very recently performed some work to show that dipole-bound states can survive in more complex environments (PCCP DOI: 10.1039/C9CP04942H), however this study was a first attempt to probe this issue. We have

included a short consideration of these states in more complex environments but refrain from making any assertive comments.

REVIEWERS' COMMENTS:

Reviewer #1 (Remarks to the Author):

The authors have made all the suggested changes in the manuscript. I therefore approve its publication as it is in Nature Communications.

Reviewer #3 (Remarks to the Author):

The authors have properly addressed the all points raised by me and the other reviewers. I recommended publication.